# Development of an LNP-Encapsulated mRNA-RBD Vaccine against SARS-CoV-2 and Its Variants

**DOI:** 10.3390/pharmaceutics14051101

**Published:** 2022-05-20

**Authors:** Cong Liu, Nino Rcheulishvili, Zhigao Shen, Dimitri Papukashvili, Fengfei Xie, Ziqian Wang, Xingyun Wang, Yunjiao He, Peng George Wang

**Affiliations:** School of Medicine, Southern University of Science and Technology, Shenzhen 518000, China; 11930759@mail.sustech.edu.cn (C.L.); nino@sustech.edu.cn (N.R.); shenzg@sustech.edu.cn (Z.S.); dimitri@sustech.edu.cn (D.P.); xieff@mail.sustech.edu.cn (F.X.); wangzq@sustech.edu.cn (Z.W.); 12032609@mail.sustech.edu.cn (X.W.)

**Keywords:** SARS-CoV-2, COVID-19, mRNA vaccine, RBD, LNPs, neutralizing antibodies, Delta, Omicron

## Abstract

Coronavirus disease 2019 (COVID-19) caused by severe acute respiratory syndrome coronavirus 2 (SARS-CoV-2) is undoubtedly the most challenging pandemic in the current century and remains a global health emergency. As the number of COVID-19 cases in the world is on the rise and variants continue to emerge, there is an urgent need for vaccines. Among all immunization approaches, mRNA vaccines have demonstrated more promising results in response to this challenge. Herein, we designed an mRNA-based vaccine encoding the receptor-binding domain (RBD) of SARS-CoV-2 encapsulated in lipid nanoparticles (LNPs). Intramuscular (i.m.) administration of the mRNA-RBD vaccine elicited broad-spectrum neutralizing antibodies and cellular responses against not only the wild-type SARS-CoV-2 virus but also Delta and Omicron variants. These results indicated that two doses of mRNA-RBD immunization conferred a strong immune response in mice against the wild-type SARS-CoV-2, while the booster dose provided a sufficient immunity against SARS-CoV-2 and its variants. Taken together, the three-dose regimen strategy of the mRNA-RBD vaccine proposed in the present study appears to be a promising reference for the development of mRNA vaccines targeting SARS-CoV-2 variants.

## 1. Introduction

Coronavirus disease 2019 (COVID-19) is disease caused by severe acute respiratory syndrome coronavirus-2 (SARS-CoV-2) [1,2]. SARS-CoV-2 is a class of single-stranded, enveloped RNA coronaviruses consisting of a spike (S), envelope (E), membrane (M), and nucleocapsid (N) proteins. Its whole genome sequence is 79.5% and 40% similar to SARS-CoV and Middle East respiratory syndrome coronavirus (MERS-CoV), respectively [3,4]. S-protein is a key protein that mediates the invasion of the host cell by the virus. When the virus invades the host, the S-protein that is protruded from the surface binds to the angiotensin-converting enzyme 2 (ACE2) on the cell, thereby mediating its entry into the cell and replication in the host body [5]. This makes S-protein the most utilized antigen target of vaccine design and development.

Remarkably, the receptor-binding domain (RBD) induces a more focused immune response and antibodies that may elicit better neutralization, which reduces the risk of antibody-dependent enhancement (ADE) of infection compared with the S-protein [6,7]. Furthermore, a number of studies also showed that the use of RBD produced high titers of neutralizing antibodies against SARS-CoV-2 infection in the absence of ADE [8,9,10,11]. Therefore, RBD represents an attractive target for designing the vaccine.

To date, according to the World Health Organization (WHO), there are currently 73 COVID-19 vaccines that have entered clinical research, and more than 180 remain in preclinical research. The main types include nucleic acid, inactivated, live-attenuated, replicating viral vector vaccines, and virus-like particles vaccines [12]. Nucleic acid vaccines comprise mRNA and DNA vaccines, which are characterized by simple preparation, safe application, and high efficiency. Since the outbreak of COVID-19, they have been brought to the fore by pharmaceutical companies. The COVID-19 vaccines, currently authorized for emergency use in many countries, include two mRNA vaccines-BNT162b2 developed by BioNTech/Pfizer [13] and mRNA-1273 by Moderna. The infection prevention effectiveness of BNT162b2 and mRNA-1273 vaccines was 95% and 94.1%, respectively [14,15]. 

Since the endosome escape efficiency of administered mRNA is low and it is easy to be cleared in the body [16,17], a suitable carrier is required to deliver it into the cell. Indeed, the main challenge for the synthetic mRNA technologies is to develop a high endosomal escape-effective and organ-specific targeting LNP delivery system. In the current mRNA vaccines, lipid nanoparticles (LNPs) are used as carriers. An optimized delivery method can not only increase the safety, stability, and convenience of the vaccine but can also enhance the vaccine outcome by improving the efficiency of cell entry and promoting the escape from endosomes. 

The SARS-CoV-2 virus tends to mutate fast. Alpha, Beta, Gamma, Delta, and Omicron are the currently designated variants of concern (VOC) by WHO while Lambda and Mu are currently identified as variants of interest [18,19]. The most critical concerns of these new variants are their transmissibility and disease severity as well as vaccine effectiveness [20,21]. Alpha, Beta, Gamma, and Delta variants increased the risk of hospitalization, intensive care unit (ICU) admission, and mortality more than the wild-type SARS-CoV-2 [22]. Compared with the Delta, a larger number of mutations were identified in the Omicron variant. These mutations give rise to markedly increased transmissibility, disease severity, and reduction in efficacy of currently available diagnostics, vaccines, and therapeutics [23,24]. Protection by the three-dose mRNA COVID-19 vaccine against SARS-CoV-2 variants in animal models has not been evaluated due to the limited time. Here, we assessed the protective efficacy of the three-dose mRNA-RBD vaccine against wild-type SARS-CoV-2, Delta, and Omicron variants, analyzed the antibody and T cell responses in mice, and importantly, our mRNA-RBD vaccine elicited robust and long-term immune responses as well as protected host cells from SARS-CoV-2, Delta, and Omicron pseudovirus infection.

## 2. Materials and Methods

### 2.1. Cells and Animals

HEK293T cells obtained from ATCC (CRL-3216) were maintained in DMEM (Gibco, Carlsbad, CA, USA) containing 10% FBS (Gibco) at 37 °C in a 5% CO_2_ incubator. Specific pathogen-free female BALB/c mice (6–8 weeks old) were purchased from Guangdong Yaokang Biotechnology Co., Ltd. (Foshan, China)

### 2.2. Construction of the EGFP, Luc, and RBD Plasmids

The pVAX1(+) vector was designed for the high yield synthesis of RNA in vitro (GENEWIZ, Suzhou, China). A tissue plasminogen activator (tPA) signal sequence was used as the signal peptide for the better expression and secretion of RBD. To confirm the effects of untranslated regions (UTRs) on the translation regulation of mRNA, the 5′ and 3′ UTRs of mRNA were derived from β-globin of *Xenopus laevis*. To test whether LNP is an effective carrier vehicle for delivering mRNA in vivo, sequence firefly luciferase (FLuc) was introduced into the vector. Enhanced green fluorescent protein (EGFP) sequence for evaluation of the optimal modification of nucleotide was employed. The concentration and purity of recombinant plasmids were automatically measured by Nanodrop 2000c (Thermo Fisher Scientific, Waltham, MA, USA).

### 2.3. In Vitro mRNA Transcription

mRNAs were produced in vitro using T7 polymerase-mediated RNA transcription. Codon-optimized open reading frames plus flanking 5′ and 3′ UTRs for all genes and a poly-A tail were synthesized by GENEWIZ. Linearized plasmid DNA was used for in vitro transcription (IVT). N1-methylpseudouridine (m1Ψ) triphosphate instead of uridine triphosphate (UTP) was used to synthesize modified nucleoside-containing mRNA. mRNA was capped at the 5′ end using a Cap1 (APExBIO, Shanghai, China) and was purified by a Monarch^®^ RNA Cleanup Kit (New England Biolabs, Ipswich, MA, USA). The concentration and purity of mRNA were measured by Nanodrop 2000c (Thermo Fisher Scientific).

### 2.4. Formulation of mRNA-LNP

LNPs were synthesized in this study by mixing one volume of lipid mixture containing an ionizable cationic lipid, phosphatidylcholine, cholesterol, and PEG–lipid at a molar ratio of 50:10:38.5:1.5. mRNA was dissolved in citrate buffer (PH 4.0). The lipid mixture and mRNA solution were mixed at a ratio of 1:3 through a T-mixer (Inano D, Micro&Nano Technology Inc., Shanghai, China). Formulations were then dialyzed against PBS (PH 7.4) through pre-sterilized Amicon^®^ Ultra-15 centrifugal filters (Millipore, Burlington, NJ, USA) to remove ethanol for 16 h and passed through a 0.22 µm filter. The mRNA-RBD LNPs were stored at 4 °C before use.

### 2.5. Particle Size, Zeta Potential, and Encapsulation Efficiency of mRNA-LNP

The size of LNPs was determined using Malvern Zetasizer instrument and visually inspected under a 300 kV cryo-electron microscopy. Zeta potential was also analyzed in PBS at pH 7.4. by the Zetasizer Nano ZS system. RNA encapsulation was calculated by the RiboGreen™ assay.

### 2.6. Transfection of Cap-mRNA into HEK293T Cells

Transfection of HEK293T cells was performed with lipofectamine 2000 (Thermo Fisher Scientific) according to the manufacturer’s instructions. mRNA-RBD (3 µg) was diluted in 250 µL Opti-MEM medium (Gibco), incubated with 3 μL of the lipofectamine 2000 for 15 min at room temperature, and the mixtures were added to the cell culture media. After 4–8 h, the medium was replaced with DMEM (Gibco). The supernatants were collected and concentrated for protein analysis.

### 2.7. Western Blot

Whole-cell lysates and supernatants from mRNA-RBD transfected cells were analyzed by Western blot. Samples were mixed with 4× loading buffer, separated in 4–20% SurePAGE, and transferred to polyvinylidene difluoride (PVDF) membrane using an eBlot^®^ L1 highly efficient wet blotting transfer system (GenScript, Nanjing, China). The membrane was blocked with 5% bovine serum albumin (BSA) in 1× phosphate-buffered saline 0.1% Tween (PBST) buffer. RBD protein was detected using 1:10,000 anti-RBD antibody for 1.5 h, and membranes were washed twice with PBST, which was followed by incubation with secondary goat anti-mouse IgG-HRP 1:10,000 (Proteintech, Manchester, UK) for 1 h. The membranes were visualized using a Tanon 5200 chemiluminescent imaging system.

### 2.8. Validation of LNPs for mRNA Delivery In Vivo

To analyze FLuc mRNA expression in tissue organs, female BALB/c mice were administered with LNPs-encapsulated Fluc mRNA via intravenous (i.v., *n* = 3), and i.m. (*n* = 3) routes at 10 µg mRNA per mice. At 6 h, 12 h, 24 h, 36 h, and 48 h after inoculation, mice were injected intraperitoneally (i.p.) with 200 µL of d-luciferin (Beyotime, Shanghai, China). After a reaction of 8 min, the mice were sacrificed, and heart, muscle, liver, kidney, spleen, and lung were collected. Bioluminescence images were acquired using the IVIS Spectrum system.

### 2.9. Mice Immunization

For immunological evaluation of mRNA-RBD in vivo, female BALB/c mice (6–8 weeks old) were immunized intramuscularly with mRNA-RBD-LNPs (low-dose 15 µg, *n* = 7; high-dose 30 µg, *n* = 7) or LNP (*n* = 7) as the placebo group. The mice were immunized with two doses and the booster dose. Serum was collected and then stored at −20 °C until the ELISA and neutralization assays. The lymphocytes in the spleen were collected on day 28 for test cell-mediated immune responses by flow cytometry. 

### 2.10. Enzyme-Linked Immunosorbent Assay

Evaluation of IgG expression was performed by ELISA. The 96-well ELISA plates (Corning, New York, NJ, USA) were coated with 2 µg/mL purified recombinant RBD protein (Sino Biological, Beijing, China) at 4 °C overnight and were blocked in 5% BSA in PBST at 37 °C for 1 h. After plates were washed twice with PBST, mouse sera were diluted and incubated on the plate for 2 h at room temperature, which was followed by three washes. The plates were incubated with secondary antibody HRP-conjugated anti-mouse IgG antibody 1:10,000 (Abcam, Cambridge, UK), which was followed by incubation with TMB substrate (Beyotime). The absorbance was measured at 450 nm by Synergy HTX (BioTeK, Winooski, VT, USA) microplate reader.

### 2.11. Flow Cytometry Analyses

Antigen-specific T cell immune responses were assayed on a multicolor flow cytometer (BD Biosciences). After collecting the splenocytes from immunized or unimmunized mice, 2 × 10^6^ cells (100 μL) per sample were stimulated with the SARS-CoV-2 S-protein peptides for 4 h at 37 °C. Brefeldin A (Thermo Scientific) was then added into splenocytes and incubated for 6 h. Stimulated cells were washed in PBS/0.5% BSA and stained with APC/Fire 750 anti-mouse CD3 antibody (BioLegend, San Diego, CA, USA), FITC anti-mouse CD4 antibody (BioLegend), and Brilliant Violet 510 anti-mouse CD8a antibody (BioLegend) surface markers. The cells were then fixed using a Fixation/Permeabilization Solution Kit (BD Biosciences) and stained with PE anti-mouse IFN-γ antibody (BioLegend), PE anti-mouse IL-2 antibody (BioLegend), and PE anti-mouse IL-4 antibody (BioLegend). Data were analyzed with FlowJo software.

### 2.12. Pseudovirus-Based Neutralization Assay

The pseudovirus-based neutralization assay was conducted as mentioned previously [25]. HEK293-ACE2 cells were seeded 20,000 cells per well in 96-well cell culture plates and incubated until 85–90% confluency. Serum samples were 3-fold diluted and mixed with pseudovirus at 37 °C for 1 h. The mixture was added to the seeded cells. After 36 h, the Fluc activity was obtained by using the Bio-Lite Luciferase Assay System (Vazyme, Nanjing, China). The percentage of neutralization was calculated, and EC50 titers were determined.

### 2.13. Statistical Analysis

All data were analyzed with GraphPad Prism version 8.0 software. For all of the analyses, data are presented as mean ± standard errors in all experiments. *p* values were determined by one-way ANOVA or *t*-test. * *p* < 0.05; ** *p* < 0.01; *** *p* < 0.001; **** *p* < 0.0001.

## 3. Results

### 3.1. Preparation and Characterization of SARS-CoV-2 mRNA-RBD Vaccine

Here, we designed an mRNA vaccine that encodes the RBD protein of SARS-CoV-2 and contains m1Ψ as a substitution of uridine (U) (Figure 1a, Appendix A). The transfection of mRNA-RBD into HEK293T cells was performed to confirm the mRNA translation abilities. As shown in Figure 1b, RBD-protein can be successfully secreted into the cell supernatant. The dynamic light scattering (DLS) of LNPs in PBS (pH 7.4) showed that the mean size was 83 nm, with more than 95% encapsulation efficiency (Figure 1c,d). Zeta potential was also measured as −10 mV at pH 7.4 (Figure 1e). Cryo-TEM analysis showed uniform size distribution of the particles (Figure 1f). In order to further explore the stability of mRNA-LNP, the particle size was measured by DLS upon storage at 4 °C, −20 °C, and −80 °C for up to 2 weeks. Remarkably, the particle size of LNPs was not changed over a 2-week duration, and the encapsulation efficiency was also stable (Figure 1g,h). These results suggested the sufficient stability of mRNA-RBD-LNPs stored at different temperatures.

To assess the tissue distribution of mRNA-LNP following administration, LNP encapsulated with mRNA-FLuc as a model molecule was prepared and then injected to BALB/c mice via i.m. and i.v. Following the i.m. administration, after 6 h, higher expression of FLuc mRNA was observed in the liver and muscle. Importantly, the Fluc level 48 h after injection was still detected at the injection site in i.m. immunized mice. The i.v. administration also elicited FLuc mRNA expression in the liver, whereas no signal was detected in mice 36 h after injection (Figure 2a). Further ex vivo imaging analysis supported the in vivo bioluminescence imaging data (Figure 2b). These results demonstrated that i.m. injection exhibited more prolonged protein expression. Thus, i.m. administration was used in the mice immunization.

### 3.2. Improvement in mRNA-RBD Expression by Modified Nucleotide

For evaluating the effects of modified nucleotides, EGFP and FLuc mRNA were synthesized using either all unmodified nucleotides or substitutions of cytidine (C) for 5-methyl-cytidine (m5C) or substitutions of U for N1-Methylpseudouridine (m1Ψ) (Figure 3a). Fluorescence microscope images revealed that mRNAs comprising m5C or m1Ψ could express EGFP. The m1Ψ-modified EGFP group had a stronger signal than the other EGFP groups (Figure 3b). Bioluminescence imaging analysis also showed that the m1Ψ-modified FLuc mRNA had a higher expression (Figure 3c). Hence, m1Ψ was utilized as the final modified nucleotide for the subsequent experiments.

### 3.3. mRNA-RBD Vaccine Induced Immune Responses In Vivo

The immune responses induced by the mRNA-RBD vaccine were assessed in mice. Groups of mice (*n* = 7) were immunized i.m. twice or three times with high (30 μg) or low (15 μg) doses. Control groups were immunized with empty LNPs. The serum samples were collected at the indicated time after initial vaccination and SARS-CoV-2 specific IgG and neutralizing antibodies were evaluated by ELISA (Figure 4a). The IgG and neutralizing antibodies were increased dramatically in mice in the low-dose and high-dose groups (Figure 4b,c), whereas no significant difference in IgG and neutralizing antibodies was observed after injection of the second dose between the low-dose and high-dose groups. Remarkably, the levels of neutralizing antibodies in low and high-dose groups remained stable at week 11, and the titers were significantly increased after the booster-dose immunization (Figure 4c).

Then, we investigated the neutralization ability of the antibodies induced by the mRNA-RBD vaccine against SARS-CoV-2 variants in vitro. The pseudovirus neutralization assays were conducted using SARS-CoV-2, Delta, and Omicron pseudoviruses. The results showed that sera from vaccinated mice were able to neutralize Delta and Omicron variants with high neutralizing activity (Figure 4d). The results demonstrated that the capacity of the candidate vaccine to neutralize the Delta variant was rapidly increased after the booster-dose immunization in mice, while the neutralizing antibody titer against the Delta variant was higher compared with the Omicron (Figure 4d). Taken together, these data suggest that three doses of the mRNA-RBD vaccine is able to provide broad protection against Delta and Omicron variants.

T cell response plays a critical role in vaccine-induced cellular immune responses to overcome the infection. The lymphocytes in the spleen of mice were collected at week 4 for evaluating the cellular immune responses. The results showed a significant increase in the percentage of CD4+ and CD8+ T cells compared with the control group (Figure 5a). Furthermore, IFN-γ production by both CD4+ and CD8+ T cells in mRNA-RBD immunized mice was significantly increased compared with vehicle treatment (Figure 5b). Therefore, our data demonstrate that the mRNA-RBD vaccine can induce cellular immune responses toward RBD protein.

## 4. Discussion

COVID-19 remains a global health crisis since its emergence in 2019 [26]. The disease manifestation varies from mild symptoms to severe acute respiratory syndrome and sometimes death [27]. Despite the fact that there are already commercially available vaccines against SARS-CoV-2, the ongoing unprecedented pandemic still necessitates an urgent need for more, safe, and effective vaccine candidates. There are various types of vaccines against SARS-CoV-2, including the conventional immunization approach. Among various immunization strategies, nucleic acid-based vaccines have certain advantages. Particularly, mRNA vaccines elicit more prominent features compared with DNA vaccines. The advantages include fast, cell-free manufacturing and production. After the delivery of mRNA, the host cell becomes a factory itself. The mRNA vaccine is carried into the cytoplasm and not in the nucleus, which eliminates the risk of genomic integration [28]. Thus, the prophylactic application of mRNA is a rapidly developing field that has a great potential of immense clinical needs [29]. The successful implementation of the current commercially available COVID-19 vaccines and the preclinical and clinical development of the other mRNA vaccines as well as mRNA therapeutics against certain non-infectious diseases is a huge warrant of the abovementioned [30,31,32]. 

In the current study, we have tested the immunogenicity and efficacy of the LNP-encapsulated, nucleoside modified mRNA-RBD vaccine against wild-type SARS-CoV-2, Delta, and Omicron variants. The results demonstrated that the immunization with two doses or three doses of mRNA-RBD elicited strong neutralizing antibody responses and provided nearly complete protection against the wild-type SARS-CoV-2, Delta, and Omicron pseudoviruses. Our data extend recent preclinical studies of the three doses of mRNA vaccine for SARS-CoV-2 variants. Compared with full-length S-protein as antigen [33,34], the mRNA vaccine used in the present study employed the RBD as an immunogen. Importantly, the strategy has been reported in previous studies to elicit a higher level of neutralizing antibodies in the absence of ADE [35,36,37,38]. Indeed, the research of mRNA-RBD is the center of foci for advancing the design of COVID-19 vaccines [37,38,39]. In accordance with these data, our results indicate that high titers of antibodies elicited by the mRNA-RBD vaccine were maintained for 2 months. 

The increasing incidence and prevalence of SARS-CoV-2 variants have influenced the effectiveness of vaccines—especially Delta and Omicron, which have become dominant variants worldwide, with remarkably enhanced pathogenicity and transmissibility, respectively [40,41,42]. The variants have the capacity to escape neutralization by serum from convalescent humans and monoclonal antibodies [43,44]. In response to variants of Delta and Omicron, our results showed that the mRNA-RBD vaccine induced a higher level of neutralizing antibodies, which can be able to efficiently neutralize Delta and Omicron pseudoviruses. The three-dose regimen used in this study can be potentially used as a reference to fight SARS-CoV-2 variants. Moreover, several advantages of the vaccine in this study deserve to be mentioned. The sucrose-containing storage buffer of mRNA-LNP in our study makes it relatively stable at 2–8 °C compared with BNT162b1 [45]. The mRNA-LNP used in this study elicited stability at 4 °C for up to 14 days. The basic structure of mRNA comprises a protein-encoding open reading frame (ORF), 5′ and 3′ UTRs, a 7-methyl guanosine 5′ cap structure, and a 3′ poly-A tail. These components can be modified to increase the stability, translation efficiency, and immune-stimulatory feature of mRNA. The 5′ UTR, 3′ UTR, and poly (A) tail profoundly influence the stability and translation of mRNA, both of which are critical concerns for vaccine development. Thus, we selected 5′ and 3′ UTRs of β-globin of *Xenopus laevis* that are found to enhance the translation and stability of mRNA [46,47]. A 100 nt-length poly-A tail is optimal for designing mRNA vaccines, as its length influences the decay of mRNA via modulating 3′ exonucleolytic degradation [48]. Hence, a 100 nucleotide-long poly-A tail was employed in this study. Additionally, tPA signal sequence was used as the signal peptide in order to enhance the RBD expression and secretion. Importantly, the modification of the 5′ Cap plays an essential role in evading the cellular innate immunity that is important to avoid the antiviral response. While Cap0 is beneficial for the translation and stability of mRNA, the Cap1 structure masks mRNA from the detection by the immune system [49,50]. For the same reason, Cap1 was employed as an optimal choice in the current study. Additionally, we used the m1Ψ instead of U, as the optimization of mRNA can significantly increase the overall efficacy of the vaccine construct. Particularly, replacement of the native nucleoside with the chemically modified nucleoside in mRNA augments the translation efficiency and modulates the stability that was also demonstrated in the present study. Along with the nucleoside modification, the improved delivery via LNP technology also modulates the mRNA vaccine efficacy [51]. Indeed, Pardi et al. have studied the effect of the nucleoside-modified mRNA vaccine in mice and demonstrated that the m1Ψ-mRNA-LNP vaccine generates strong T cell responses, creates durable neutralizing antibody responses, and induces the long-term production of antigens [52]. The LNP formulated in this study could escape the endosomal membranes, deliver the mRNA construct, and release it into the cytoplasm. Further results demonstrated the capacity of our mRNA packed in LNP to recruit antigen-presenting cells to process the RBD antigens. 

Taken together, our study demonstrated that the mRNA-RBD vaccine protected host cells from SARS-CoV-2, Delta, and Omicron pseudovirus infection after three doses by inducing cellular immune responses. Nevertheless, the limitations of the study deserve to be mentioned. Albeit the proposed vaccine has demonstrated good protection against SARS-CoV-2 and its variants in mice, its safety and efficacy have not been further evaluated in non-human primates or humans.

## 5. Conclusions

To the best of our knowledge, this is the first study that used the mRNA-based vaccine expressing SARS-CoV-2 RBD immunization three times against Delta and Omicron variants. In summary, the mRNA-RBD vaccine designed and validated in the current study evidenced sufficient cellular immune responses and protection in mice. The proposed vaccine seems efficient enough to be considered as a promising candidate against SARS-CoV-2 and its variants. Further research including the viral challenge and the studies of non-human primates and humans are necessary.

## Figures and Tables

**Figure 1 pharmaceutics-14-01101-f001:**
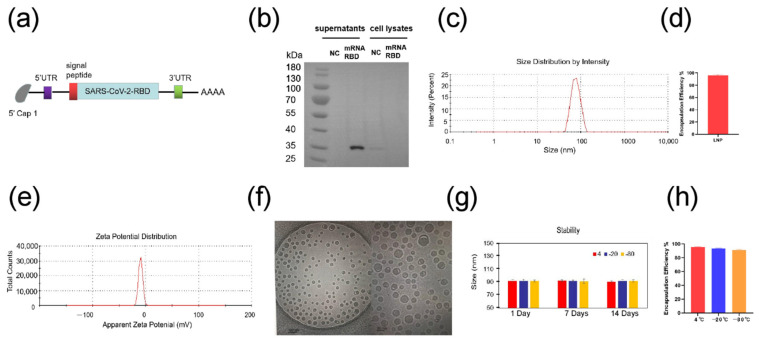
Preparation and characterization of SARS-CoV-2 mRNA-RBD vaccine. (**a**) The construct of mRNA-RBD vaccine for COVID-19. (**b**) The expression of RBD mRNA in HEK293T cells (**c**) The size distribution of LNPs was measured by a Malvern particle size instrument. (**d**) The encapsulation efficiency of LNPs using the ribogreen assay (**e**) The zeta potential for LNPs at pH 7.4 PBS. (**f**) Cryo-TEM images of the LNPs (scale bar, 100 nm, 50 nm). (**g**) Time-dependent change in particle size of LNPs upon storage at 4 °C, −20 °C and −80 °C. (**h**) Time-dependent change in encapsulation efficiency of LNPs for 14 days using the ribogreen assay upon storage at 4 °C, −20 °C and −80 °C. UTR, untranslated region; NC, negative control.

**Figure 2 pharmaceutics-14-01101-f002:**
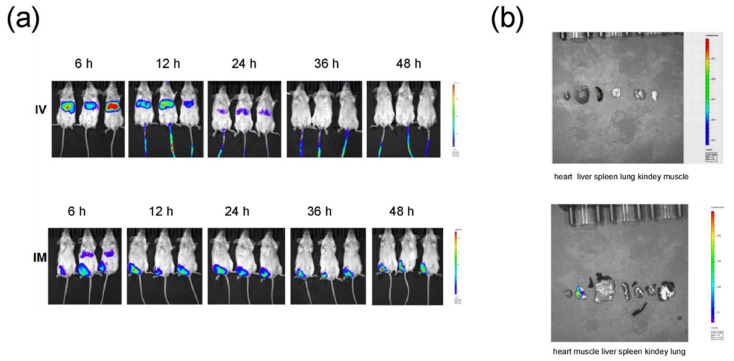
In vivo delivery of mRNA-LNPs formulation. (**a**) Duration and expression pattern of LNPs-encapsulated Fluc mRNA in mice injected by i.v. and i.m. routes. BALB/c mice were injected with 10 µg of FLuc mRNA encapsulated by LNPs by the intramuscular (i.m.) and intravenous (i.v.) routes and representative IVIS images. (**b**) Representative Fluc expression in different organs for FLuc mRNA-LNP injected mice under IVIS imaging.

**Figure 3 pharmaceutics-14-01101-f003:**
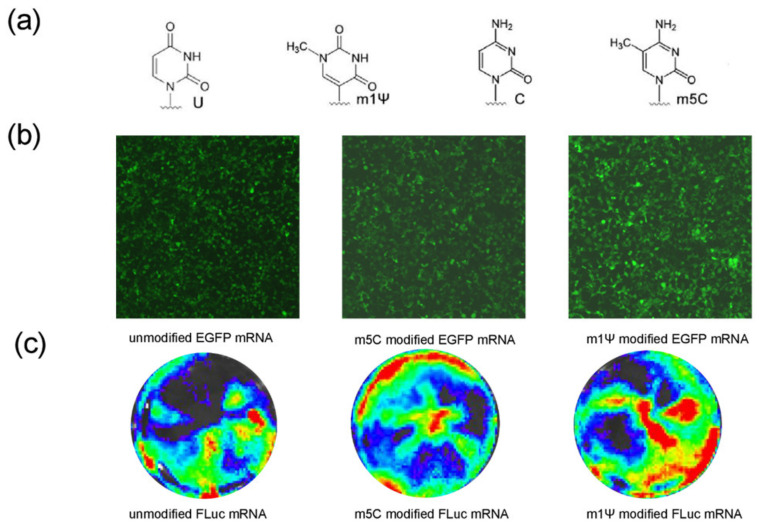
m1Ψ substitution of U improves mRNA expression. (**a**) Chemical structures of the U, C, and 2 modified nucleobase (m1Ψ, m5C). (**b**) Fluorescence microscope analyses of modified EGFP mRNAs harboring U, m5C, and m1Ψ translated in HEK293T cells. (**c**) IVIS analyses of modified FLuc mRNAs harboring U, m5C, and m1Ψ translated in HEK293T cells. m1Ψ, N1-methylpseudouridine; U, uridine; C, cytidine; EGFP, enhanced green fluorescent protein; FLuc, firefly luciferase.

**Figure 4 pharmaceutics-14-01101-f004:**
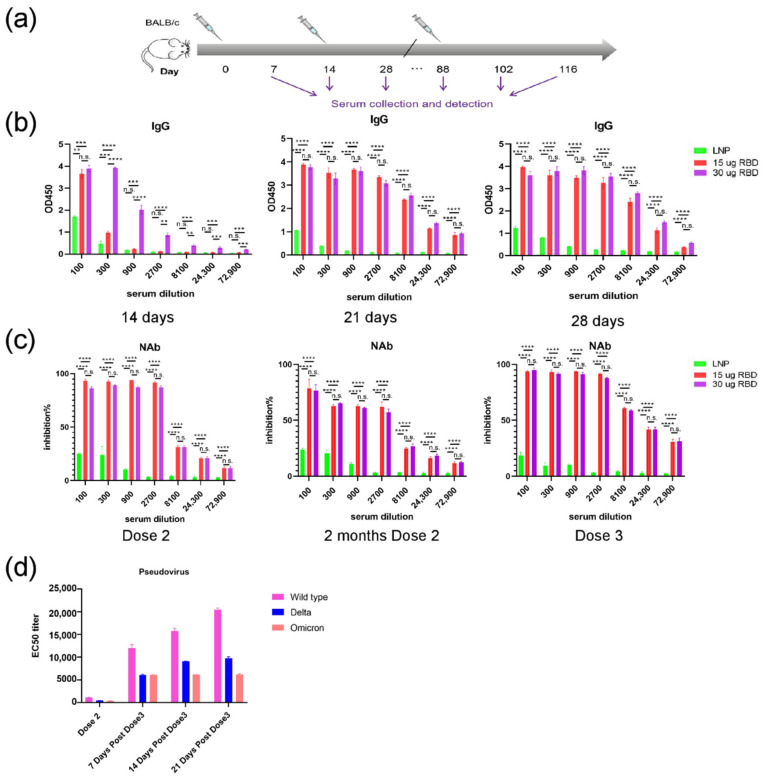
Humoral immune response in mRNA-RBD vaccinated mice. (**a**) Mice immunization and sample collection schedule (**b**) SARS-CoV-2 RBD-specific IgG concentrations were assayed by ELISA. (**c**) Inhibition of RBD-hACE2 interaction by sera from mRNA-RBD vaccinated mice. (**d**) Pseudovirus neutralization assay of the mRNA-RBD group shows the EC50 titers for the SARS-CoV-2, Delta, and Omicron pseudovirus. NAb, neutralizing antibodies; IgG, immunoglobulin G. (** *p* < 0.01; *** *p* < 0.001; **** *p* < 0.0001, n.s. no significance.)

**Figure 5 pharmaceutics-14-01101-f005:**
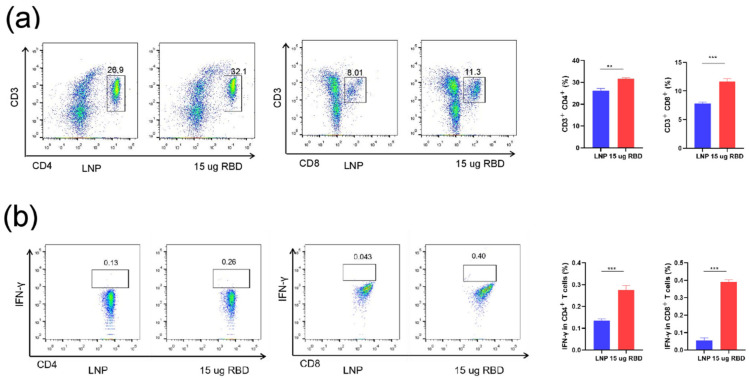
Cell immune response in mice by mRNA-RBD vaccines. (**a**) The proportions of CD3+CD4+ and CD3+CD8+ T cells were tested by flow cytometry. (**b**) The percentages of IFNγ-producing CD4+ and CD8+ T cells were detected by intracellular cytokine staining. LNP, lipid nanoparticle; RBD, receptor-binding domain. (** *p* < 0.01; *** *p* < 0.001).

## Data Availability

Not applicable.

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
