# Peer review of "Development of an LNP-Encapsulated mRNA-RBD Vaccine against SARS-CoV-2 and Its Variants"

_pharmaceutics, 2022, doi:10.3390/pharmaceutics14051101_

Round 1
Reviewer 1 Report
In this manuscript, Liu et al. developed and evaluated a lipid nanoparticle encapsulated mRNA-RBD vaccine in a mouse model. The results indicated that the candidate vaccine elicited cellular immune responses and protection in mice. Two doses of the candidate vaccine strong immune responses against SARS-CoV2 and the inclusion of additional booster dose provided protection against Delta and Omicron pseudovirus infection. The authors acknowledge that although the candidate vaccine may hold promise, it would require further evaluation with regards to safety and efficacy in non-human primates or humans. I have only a few minor comments.
- Abstract, line 16: Change “responded” to in response
- Introduction, line 342: The authors refer to gene sequence similarity, but this likely to comparisons of viral genomes sequences. Please correct accordingly.
- How many mice were used in each group? Under Material and Methods it is indicated that the groups consisted of n=7 (Page 3, Line 143), but under Results it is indicated as n=6 (Page 6, line 235).
Author Response
Dear reviewer,
We would like to thank you for reviewing our manuscript (Manuscript ID: 1630686) entitled “Development of an LNP-Encapsulated mRNA-RBD Vaccine Against SARS-CoV-2 and Its Variants.” We appreciate your feedback, useful comments, and suggestions.
In the manuscript, we have used the “Track Changes” function for all the modifications. Additionally, we have corrected some misspelled errors to improve our manuscript. Please find below a point-by-point reply. We hope you will find our responses satisfactory.
Sincerely,
Authors
Comments and Suggestions for Authors:
In this manuscript, Liu et al. developed and evaluated a lipid nanoparticle encapsulated mRNA-RBD vaccine in a mouse model. The results indicated that the candidate vaccine elicited cellular immune responses and protection in mice. Two doses of the candidate vaccine strong immune responses against SARS-CoV2 and the inclusion of additional booster dose provided protection against Delta and Omicron pseudovirus infection. The authors acknowledge that although the candidate vaccine may hold promise, it would require further evaluation with regards to safety and efficacy in non-human primates or humans. I have only a few minor comments.
Response: First of all, we are thankful for the positive comment and suggestions.
Comment 1:
Abstract, line 16: Change “responded” to in response
Response: The word “respond” has been replaced by “in response” in the sentence (abstract, line 16).
Comment 2:
Introduction, line 342: The authors refer to gene sequence similarity, but this likely to comparisons of viral genomes sequences. Please correct accordingly.
Response: We are grateful for pointing out this error. “Their gene sequence” has been replaced by “Its whole genome sequence” in the sentence as follows (line 33):
“Its whole genome sequence is 79.5% and 40% similar to SARS-CoV and Middle East respiratory syndrome coronavirus (MERS-CoV), respectively [2,3].”
Comment 3:
How many mice were used in each group? Under Material and Methods it is indicated that the groups consisted of n=7 (Page 3, Line 143), but under Results it is indicated as n=6 (Page 6, line 235).
Response: We apologize for this oversight. In this study, 7 mice were used in each group. We have corrected this mechanical error (line 245).

Reviewer 2 Report
Dear Authors
This paper is written well but there are a few concerns to strengthen this manuscript.
Abstract is good but the keywords are quite weak. Authors have to add more and appropriate so this paper will visible to all readers.
Add these papers information in first paragraph of the paper, it may give good start on the main topic.
1) Hamid, Hiba, et al. "COVID-19 pandemic and role of human saliva as a testing biofluid in point-of-care technology." European journal of dentistry 14.S 01 (2020): S123-S129.
2) Khurshid, Z., Asiri, F. Y. I., & Al Wadaani, H. (2020). Human saliva: non-invasive fluid for detecting novel coronavirus (2019-nCoV). International journal of environmental research and public health, 17(7), 2225.
Result part and method part are well written. No need of editing.
Regarding Discussion, I would recommend authors to double-check if any study linked with their work try to compare with the current study outcomes.
The COnclusion needs serious attention to readers can get benefit as well as allow to work on the gaps reported by this study.
Author Response
Dear reviewer,
We would like to thank you for reviewing our manuscript (Manuscript ID: 1630686) entitled “Development of an LNP-Encapsulated mRNA-RBD Vaccine Against SARS-CoV-2 and Its Variants.” We appreciate your feedback, useful comments, and suggestions.
In the manuscript, we have used the “Track Changes” function for all the modifications. Additionally, we have corrected some misspelled errors to improve our manuscript. Please find below a point-by-point reply. We hope you will find our responses satisfactory.
Sincerely,
Authors
Comments and Suggestions for Authors:
Dear Authors
This paper is written well but there are a few concerns to strengthen this manuscript.
Response: We would like to express our gratitude for the complimentary comments and helpful suggestions.
Comment 1:
Abstract is good but the keywords are quite weak. Authors have to add more and appropriate so this paper will visible to all readers.
Response: We appreciate the positive comment. We have taken the suggestion into consideration and added more keywords “SARS-CoV-2”, “neutralizing antibodies”. Please see line 26.
Comment 2:
Add these papers information in first paragraph of the paper, it may give good start on the main topic.
1) Hamid, Hiba, et al. "COVID-19 pandemic and role of human saliva as a testing biofluid in point-of-care technology."European journal of dentistry14.S 01 (2020): S123-S129.
2) Khurshid, Z., Asiri, F. Y. I., & Al Wadaani, H. (2020). Human saliva: non-invasive fluid for detecting novel coronavirus (2019-nCoV).International journal of environmental research and public health,17(7), 2225.
Response: We have considered your suggestion and cited these papers in the introduction and discussion parts (line 31 and lines 286-288).
Comment 3:
Result part and method part are well written. No need of editing.
Regarding Discussion, I would recommend authors to double-check if any study linked with their work try to compare with the current study outcomes.
The Conclusion needs serious attention to readers can get benefit as well as allow to work on the gaps reported by this study.
Response: First of all, we appreciate the positive comment regarding the Method and Result parts. According to the suggestion, we have cited the following paper in the Discussion (lines 311-312):
Huang, Q., Ji, K., Tian, S. et al. A single-dose mRNA vaccine provides a long-term protection for hACE2 transgenic mice from SARS-CoV-2. Nat Commun 12, 776 (2021).
However, we have not compared our study to the cited paper in detail because of certain reasons, e.g., the absence of Delta and Omicron variants during the study while our study includes the mentioned variants of SARS-CoV-2.
Besides, we consider that the Conclusion already corresponds to the raised points as we have already mentioned the benefit as well as the main limitations of the study.

Reviewer 3 Report
This article details the application of RBD enconded mRNA-LNPs against SARS-COV2. Clear representation of experiments and results. However, few minor suggestions are required to improve the manuscript quality.
- A high-resolution TEM image with 10 nm scale bar would help to get clear understanding on the morphology of LNPs.
- Experimental conditions how the LNPs were stored at -80C should be mentioned. Also, compared to show the size, it would be good to see the expression levels from the LNPs after different time points and different storage conditions.
- Figure 2: As a control, authors should have injected just FLuc mRNA and compared with FLuc-LNPs to confirm the expression and support the rationale for using LNPs.
- The background noise from m5C modified EGFP mRNA is higher. Please reduce the background or make all the experimental analysis conditions similar to compare the expression.
- In Figure 5, authors should have used RBD mRNA treated animals without LNPs or just control mice without any injections as control to show an increase in the CD4/8 and activated T-cell expression.
- The gating could be changed to show a difference in the increased IFN-gamma positive T-cells. It would be clear representation if the authors could sum the data in bar plots to determine the mean enhanced activation of T-cells between groups and animals.
Author Response
Dear reviewer,
We would like to thank you for reviewing our manuscript (Manuscript ID: 1630686) entitled “Development of an LNP-Encapsulated mRNA-RBD Vaccine Against SARS-CoV-2 and Its Variants.” We appreciate your feedback, useful comments, and suggestions.
In the manuscript, we have used the “Track Changes” function for all the modifications. Additionally, we have corrected some misspelled errors to improve our manuscript. Please find below a point-by-point reply. We hope you will find our responses satisfactory.
Sincerely,
Authors
Comments and Suggestions for Authors:
This article details the application of RBD enconded mRNA-LNPs against SARS-COV2. Clear representation of experiments and results. However, few minor suggestions are required to improve the manuscript quality.
Response: We are grateful for the positive evaluation and the valuable suggestions.
Comment 1:
A high-resolution TEM image with 10 nm scale bar would help to get clear understanding on the morphology of LNPs.
Response: We appreciate pointing out this error. However, we have included the TME image with 50 nm scale bar in figure 1(f) as we only have a high-resolution TEM image with 50 nm scale bar because of the software fault. The purpose of using 100 nm is to clearly observe whether LNP is uniformly distributed. If we have used 10nm, we could only see the morphology of LNP more clearly, but not all LNP distributions. Please see the graph below. The figure 1 legend has been also accordingly modified.
Comment 2:
Experimental conditions how the LNPs were stored at -80C should be mentioned. Also, compared to show the size, it would be good to see the expression levels from the LNPs after different time points and different storage conditions.
Response: We are grateful for making this important comment. For storage of LNPs at -80 ℃ sucrose was used as a cryoprotectant. As to the expression levels, we only tested the encapsulation efficiency and particle size of LNP at different temperatures. As particle size and encapsulation efficiency did not change obviously, we assumed that the expression level of the coded gene would not be affected. Please see Figure 1(h).
Comment 3:
Figure 2: As a control, authors should have injected just FLuc mRNA and compared with FLuc-LNPs to confirm the expression and support the rationale for using LNPs.
Response: Although FLuc mRNA can be expressed by injecting naked FLuc mRNA in animal, it is quickly degraded by extracellular RNases and is not internalized efficiently, thus, the level of expression is usually low. Therefore, we have used Fluc only to test whether LNP was an effective delivery system in vivo.
Comment 4:
The background noise from m5C modified EGFP mRNA is higher. Please reduce the background or make all the experimental analysis conditions similar to compare the expression.
Response: According to the suggestion, we have reduced the background in a graph of m5C modified EGFP.
Comment 5:
In Figure 5, authors should have used RBD mRNA treated animals without LNPs or just control mice without any injections as control to show an increase in the CD4/8 and activated T-cell expression.
Response: Since the purpose of this study is to evaluate the immune respose induced by mRNA vaccine which encodes the RBD protein, so the blank LNP without any mRNA is used as a control. If the naked mRNA-RBD used as a control, the immune response induced by the encoded RBD is unignorable. If the untreated mice is used as the control, the immune response induced by the LNP can not be excluded. Based on the principle of controlled experiments, there is only one variable between the control group and the experimental group. So our control group was treated with LNP, while experimental group was treated with mRNA RBD encapsulated via LNP. The same principle was used by Huang et al. 2021 (doi.org/10.1038/s41467-021-21037-2)
Comment 6:
The gating could be changed to show a difference in the increased IFN-gamma positive T-cells. It would be clear representation if the authors could sum the data in bar plots to determine the mean enhanced activation of T-cells between groups and animals.
Response: We thank the reviewer for the suggestion. We have modified Figure 5 and sum the data in bar plots accordingly.

Reviewer 4 Report
My comments:
- LNP preparation method need more details information?
- What is the reason behind more stable at 4 degrees where as Moderna or Pfizer vaccines are not?
- In Figure 1, Y-axis scale bar need to adjust in stability graph.

Author Response
Dear reviewer,
We would like to thank you for reviewing our manuscript (Manuscript ID: 1630686) entitled “Development of an LNP-Encapsulated mRNA-RBD Vaccine Against SARS-CoV-2 and Its Variants.” We appreciate your feedback, useful comments, and suggestions.
In the manuscript, we have used the “Track Changes” function for all the modifications. Additionally, we have corrected some misspelled errors to improve our manuscript. Please find below a point-by-point reply. We hope you will find our responses satisfactory.
Sincerely,
Authors
Comments and Suggestions for Authors:
Comment 1:
LNP preparation method need more details information?
Response: We apologize for explaining the preparation of LNP not clearly enough. LNP nanoparticles were prepared by a two-step method. An organic solvent containing dissolved lipids and an aqueous solution containing mRNA are injected into the two inlet channels of the chip. Under laminar flow, the two solutions do not immediately mix, but microscopic features engineered into the channel cause the two fluids to intermingle in a controlled and reproducible way, where molecules interact with each other by diffusion. Within 1 millisecond, the two fluids are completely mixed, causing a change in solvent polarity that triggers the homogenous self-assembly of LNP loaded with mRNA. Please see lines 109-115.
LNPs were synthesized in this study by mixing one volume of lipid mixture con-taining an ionizable cationic lipid, phosphatidylcholine, cholesterol, and PEG-lipid at a molar ratio of 50:10:38.5:1.5. mRNA was dissolved in citrate buffer (PH 4.0). The lipid mixture and mRNA solution were mixed at a ratio of 1:3 through a T-mixer (Inano D, Micro&Nano Technology Inc, China). Formulations were then dialyzed against PBS (PH 7.4) through pre-sterilized Amicon® Ultra-15 centrifugal filters (Millipore) to re-move ethanol for 16 h and passed through a 0.22 µm filter. The mRNA-RBD LNPs were stored at 4 ℃ before use.
Comment 2:
What is the reason behind more stable at 4 degrees where as Moderna or Pfizer vaccines are not?
Response: We are grateful for pointing out this critical issue.
Actually, most laboratory-made LNPs are stable at 4 °C for several days, but then exhibit size increase and progressive loss of biological activity. It has been reported that the Moderna COVID-19 vaccine is stable for 30 days at 2 °C~8 °C, and the Pfizer/BioNTech COV1D-19 vaccine can be stored at 2 °C to 8 °C for up to 5 days Please refer to the table below (Source: Crommelin et al. 2021 (doi.org/10.1016/j.xphs.2020.12.006)). The main reason for the difference in stability between these two is that Moderna's vaccine is stored in a buffer containing high concentrations of sucrose, while Pfizer-BioNTech's vaccine is stored in PBS, which suggesting the cryoprotectant sucrose is critical for the mRNA-LNP stability.
Comment 3:
In Figure 1, Y-axis scale bar need to adjust in stability graph.
Response: We thank the reviewer for this important comment. We have adjusted the graph accordingly.

Reviewer 5 Report
The manuscript entitled “Development of an LNP-Encapsulated mRNA-RBD Vaccine Against SARS-CoV-2 and Its Variants” present the results about in vitro and in vivo with a lipid-based nanoparticle formulation containing mRNA coding for the RBD domain of SARS-CoV-2 spike. There are minor issues (English, lack of info as exemplified below) but a major issue is that this work does not bring anything new versus BNT162b1. Thus, it is not clear for me why those results should be published. What would be the novelty (mRNA design or formulation) versus BNT162b1? Is there something in the present formulation that would make it better than BNT162b1?
Minor issues:
16 “more promising results respond to this challenge” What does it mean?
39 “(RBD) induces the specific antibodies blocks RBD recognition by the receptor” What does it mean?
50 “ long-lasting immunity”, no the mRNA vaccines actually seem so far to induce immunity lasting only a few months.
51 “have been favored by… Pfizer and Moderna”. The technology is BioNTech, not Pfizer. BioNTech an Moderna are specialised on mRNA and did not chose it after the outbreak of COVID-19
56 “the endocytosis efficiency of nucleic acids is low and it is easier to be cleared in the body”. What does it mean?
57 “lack of delivery systems is currently the main obstacle”, no, LNPs in BioNTech/Pfizer and Moderna vaccines are working very well.
2.2 What is the leader sequence that was put in from of RBD?
99 “using Cap1 (APExBIO). On the webpage of the company there is only ARCA (a Cap0) that is offered not CleanCap (Cap1)
Figure 1: Percentage encapsulation is not shown although the Material&Methods describe RiboGreen test. Could this be shown? It should also be shown for stability studies.
Figure 2: Is the mRNA used here modified (m1PseudoU)?
222 “That two modified nucleotides could express EGFP” What does that mean?
235 “low dose” 15 micrograms in a 20 grams mouse is a big dose (it is 30 micrograms in a human!)
240 “no significant difference”. It does not look to be the case at day 14 (300 and 900 and 2700 serum dilutions)
Figure 4: What is the number of mice per group
Figure 5. Why are CD4 or CD8 cells CD3 negative?
318 “improves the immune response”, no, this study does not show that modified nucleotides improve the immune response.
Author Response
Dear reviewer,
We would like to thank you for reviewing our manuscript (Manuscript ID: 1630686) entitled “Development of an LNP-Encapsulated mRNA-RBD Vaccine Against SARS-CoV-2 and Its Variants.” We appreciate your feedback, useful comments, and suggestions.
In the manuscript, we have used the “Track Changes” function for all the modifications. Additionally, we have corrected some misspelled errors to improve our manuscript. Please find below a point-by-point reply. We hope you will find our responses satisfactory.
Sincerely,
Authors
Comments and Suggestions for Authors:
The manuscript entitled “Development of an LNP-Encapsulated mRNA-RBD Vaccine Against SARS-CoV-2 and Its Variants” present the results about in vitro and in vivo with a lipid-based nanoparticle formulation containing mRNA coding for the RBD domain of SARS-CoV-2 spike. There are minor issues (English, lack of info as exemplified below) but a major issue is that this work does not bring anything new versus BNT162b1. Thus, it is not clear for me why those results should be published. What would be the novelty (mRNA design or formulation) versus BNT162b1? Is there something in the present formulation that would make it better than BNT162b1?
Response: We appreciate the comment regarding minor issues that helped us to make the overall improvement in the manuscript. We are providing the point-by-point reply for the minor issues that are given below. But first, we would like to clarify the critical concern raised in the comment regarding the lack of novelty compared to the BNT162b1 vaccine.
Firstly, the sucrose containing storage buffer of mRNA-LNP in our study makes it more stable than BNT162b1 at 2-8°C. Secondly, the components of LNP in our study differs from the LNP of Pfizer-BioNTech. The stability of BNT162b1 at 2–8 °C is up to 5 days. Our mRNA-LNP was stable at 4°C for up to 14 days. Most importantly, our study indicates the third boost immunization maybe more helpful in fighting against Delta variant strain than against Omicron.
The basic structure of mRNA comprises (i) protein encoding open reading frame (ORF), (ii) 5′ and 3′ UTRs, (iii) a 7-methyl guanosine 5′ cap structure, and (iv) a 3′poly(A) tail. These components can be modified or altered to increase the stability, translation efficiency, and immune-stimulatory feature of mRNA. The 5′ UTR, 3′ UTR, and poly (A) tail elements profoundly influence the stability and translation of mRNA, both of which are critical concerns for vaccines.
Compared to BNT162b1, we selected 5' and 3' UTRs of β‐globin of Xenopus laevis.
5' and 3' UTRs of β‐globin of Xenopus laevis are found to enhance the translation and stability of mRNA in many studies (Linares-Fernández et al. 2021 (doi.org/10.1016/j.omtn.2021.10.007); Schlake et al. 2012 (doi.org/10.4161/rna.22269)). Poly(A) tails of 100 nt are ideal for designing mRNA vaccines because poly(A) tail length affects mRNA decay by modulating 3′ exonucleolytic degradation. So we selected 100 nucleotide-long poly(A) tail. Additionally, we selected the tissue plasminogen activator (tPA) signal sequence as the signal peptide. We introduced the tPA signal sequence in order to increase the RBD antigen expression and secretion.
Minor issues
Comment 1:
16 “more promising results respond to this challenge” What does it mean?
Response: We have modified the sentence as follows (please see line 16):
Among all immunization approaches, mRNA vaccines have demonstrated more promising results in response to this challenge.
Comment 2:
39 “(RBD) induces the specific antibodies blocks RBD recognition by the receptor” What does it mean?
Response: We apologize for the ambiguity caused. Antibody-dependent enhancement (ADE) is a concern of vaccine development as it has the potential of enhancing viral infection. There is more chance of ADE in the Spike-protein-based vaccine compared with the RBD vaccine. RBD can focus the immune response on the interference of receptor binding and theoretically entails a lower risk of inducing antibodies that readily mediate ADE of infection compared with the S-protein. To make our sentence clearer for the readers we have deleted “blocks RBD recognition by the receptor” and revised the sentence as follows (lines 40-42):
Remarkably, the receptor-binding domain (RBD) induces the specific antibodies and reduces the risk of antibody-dependent enhancement (ADE) of infection compared with the S-protein.
Comment 3:
50 “ long-lasting immunity”, no the mRNA vaccines actually seem so far to induce immunity lasting only a few months.
Response: We have modified the sentence as follows (lines 50-51):
Nucleic acid vaccines comprise mRNA and DNA vaccines, which are characterized by simple preparation, safe application, and high efficiency.
Comment 4:
51 “have been favored by… Pfizer and Moderna”. The technology is BioNTech, not Pfizer. BioNTech an Moderna are specialised on mRNA and did not chose it after the outbreak of COVID-19
Response: We have corrected this inaccuracy (lines 51-54).
Comment 5:
56 “the endocytosis efficiency of nucleic acids is low and it is easier to be cleared in the body”. What does it mean?
Response: We apologize for being not clear enough about this point. mRNA is a single-stranded RNA molecule, thus, naked mRNA is quickly degraded by extracellular RNases and is not internalized efficiently. Hence, LNPs have been developed that facilitate cellular uptake of mRNA and protect it from degradation. LNPs allow mRNA to escape from the endosomal lumen. To make the mentioned sentence more understandable, we have modified it accordingly (lines 58-59):
Since the endosome escape efficiency of mRNA is low and it is easy to be cleared in the body.
Comment 6:
57 “lack of delivery systems is currently the main obstacle”, no, LNPs in BioNTech/Pfizer and Moderna vaccines are working very well.
Response: Although LNPs in BioNTech/Pfizer and Moderna vaccines are working very well, their stability is different. LNP of Moderna can be stored at 4°C or -20°C. LNP of BioNTech can be stored at -80°C. Currently, it is impossible LNPs of BioNTech/Pfizer and Moderna vaccines to selectively target specific tissues. In recent years, LNPs have been developed to overcome the limitations of free therapeutics and navigate and overcome biological barriers. However, LNPs can still be limited by low drug loading and biodistribution which results in high uptake by the liver and spleen. To make the point clearer, we have revised the sentence in the manuscript as follows (lines 60-62):
Indeed, the main challenge for the mRNA vaccine is to develop a high endosomal escape-effective and organ-specific targeting LNP delivery system.
Comment 7:
2.2 What is the leader sequence that was put in from of RBD?
Response: We have selected the tissue plasminogen activator (tPA) signal sequence as the signal peptide in order to increase the RBD antigen expression and secretion. The tPA signal sequence is efficient in facilitating the transport of RBD protein from the endoplasmic reticulum to the Golgi apparatus. We have also added this information in the manuscript (lines 91-92):
Tissue plasminogen activator (tPA) signal sequence was used as the signal peptide for the better expression and secretion of RBD.
Comment 8:
99 “using Cap1 (APExBIO). On the webpage of the company there is only ARCA (a Cap0) that is offered not CleanCap (Cap1)
Response: On the webpage of the company, there have an EZ Cap (Cap1). Please see the below screenshot.
Comment 9:
Figure 1: Percentage encapsulation is not shown although the Material&Methods describe RiboGreen test. Could this be shown? It should also be shown for stability studies.
Response: It is a great point. We have made added the suggested information. Please see Figure 1.
Comment 10:
Figure 2: Is the mRNA used here modified (m1PseudoU)?
Response: Yes, we used N1-Methylpseudouridine (m1Ψ) in our study. m1Ψ can enhance immune evasion and protein production. The important aspect is that m1Ψ reduces mRNA immunogenicity.
Comment 11:
222 “That two modified nucleotides could express EGFP” What does that mean?
Response: In our study, for evaluating the effects of two types of modified nucleotides, m5C and m1Ψ were used. EGFP was synthesized using either all unmodified nucleotides or substitutions of cytidine (C) for 5-methyl-cytidine (m5C) or substitutions of U for m1Ψ. We have also modified a sentence to make the point clearer (lines 230-231):
Fluorescence microscope images revealed that mRNAs comprising m5C or m1Ψ could express EGFP.
Comment 12:
235 “low dose” 15 micrograms in a 20 grams mouse is a big dose (it is 30 micrograms in a human!)
Response: We appreciate your comment that has indicated a reasonable fact. We have used 15 micrograms as a low dose because we needed to consider the endosomal escape efficiency of LNP-mRNA. Endosomal escape of LNP-mRNA depends on the molar ratio between ionizable lipids and mRNA nucleotides. Besides, some studies have also used the doses of as high as 20 µg (Zhang et al. 2020, doi.org/10.1016/j.omtm.2020.07.013), 30 µg (Pardi et al. 2018, doi.org/10.1084/jem.20171450), or even 80 µg (Schnee et al. 2016, doi:10.1371/journal.pntd.0004746) mRNA for the immunization of mice.
Comment 13:
240 “no significant difference”. It does not look to be the case at day 14 (300 and 900 and 2700 serum dilutions)
Response: We have made the correction accordingly (lines 250-251):
…no significant difference in IgG and neutralizing antibodies was observed after injection of the second dose between the low-dose and high-dose groups.
Comment 14:
Figure 4: What is the number of mice per group
Response: We used 7 mice in each group. We have corrected the mechanical error in the Results section. Please see line 245.
Comment 15:
Figure 5. Why are CD4 or CD8 cells CD3 negative?
Response: CD3+CD4+cells and CD3+CD8+cells are positive. The left subgroup is negative, and the right subgroup is positive. Because in the graph, the Y-axis is not logarithmic scaling.
Comment 16:
318 “improves the immune response”, no, this study does not show that modified nucleotides improve the immune response.
Response: We have modified the sentence as follows (lines 331-332):
Particularly, replacement of the native nucleoside with the chemically modified nucleoside in mRNA augments the translation efficiency and modulates the stability that was also demonstrated in the present study.

Round 2
Reviewer 2 Report
Dear Authors
paper is well revised.
Author Response
Dear reviewer,
We would like to thank you for your positive feedback about our manuscript (Manuscript ID: 1630686) entitled “Development of an LNP-Encapsulated mRNA-RBD Vaccine Against SARS-CoV-2 and Its Variants.”
Sincerely,
Authors
Reviewer 5 Report
Thanks for the corrections. I still have minor issues and recommendations as listed below:
Line 40: «Remarkably, the receptor-binding domain (RBD) induces the specific antibodies». No. The whole spike induces specific antibodies. Those against RBD may give the best neutralization
Line 49: Ref 12 and 13 are from 2020. Maybe a more recent review on vaccines against COVID-19 can be cited as for example MDPI Cells Oct 2021 https://pubmed.ncbi.nlm.nih.gov/34685696/
Line 58 “Since the endosome escape efficiency of administered mRNA”
Line 60 “for the mRNA vaccine is ” “for the synthetic mRNA technologies is ”
Line 63: “A suitable deliv-” “An optimized deliv-”
In the introduction is missing a line describing BNT162b1 and reference Dec 2020 https://pubmed.ncbi.nlm.nih.gov/33053279/
Line 91: could the full mRNA sequence (and the resulting protein sequence) be disclosed in a supporting figure?
Figure 1H: how long was the formulation stored at those temperatures in 1h?
Line 224. It should be clearly said here that it is N1-Methylpseudouridine (m1Ψ) i
Line 236 “Hence, m1Ψ triphosphate was” “Hence, m1Ψ was”
Figure 5: I do not understand in A that CD4 and CD8 pos cells are not clearly CD3 positive. Or we do not see here CD3 negative cells in the graph (and they are CD3 positives although they are in the lower part of the dot plots?)?
In the discussion (for example line 314) the differences between BNT162b1 and the present work should be mentioned as did the authors in their answer “Firstly, the sucrose containing storage buffer of mRNA-LNP in our study makes it more stable than BNT162b1 at 2-8°C. Secondly, the components of LNP in our study differs from the LNP of Pfizer-BioNTech. The stability of BNT162b1 at 2–8 °C is up to 5 days. Our mRNA-LNP was stable at 4°C for up to 14 days. Most importantly, our study indicates the third boost immunization maybe more helpful in fighting against Delta variant strain than against Omicron. The basic structure of mRNA comprises (i) protein encoding open reading frame (ORF), (ii) 5′ and 3′ UTRs, (iii) a 7-methyl guanosine 5′ cap structure, and (iv) a 3′poly(A) tail. These components can be modified or altered to increase the stability, translation efficiency, and immune-stimulatory feature of mRNA. The 5′ UTR, 3′ UTR, and poly (A) tail elements profoundly influence the stability and translation of mRNA, both of which are critical concerns for vaccines. Compared to BNT162b1, we selected 5' and 3' UTRs of β‐globin of Xenopus laevis. 5' and 3' UTRs of β‐globin of Xenopus laevis are found to enhance the translation and stability of mRNA in many studies (Linares-Fernández et al. 2021 (doi.org/10.1016/j.omtn.2021.10.007); Schlake et al. 2012 (doi.org/10.4161/rna.22269)). Poly(A) tails of 100 nt are ideal for designing mRNA vaccines because poly(A) tail length affects mRNA decay by modulating 3′ exonucleolytic degradation. So we selected 100 nucleotide-long poly(A) tail. Additionally, we selected the tissue plasminogen activator (tPA) signal sequence as the signal peptide. We introduced the tPA signal sequence in order to increase the RBD antigen expression and secretion.”
Author Response
Dear reviewer,
We would like to thank you for your feedback about our manuscript (Manuscript ID: 1630686) entitled “Development of an LNP-Encapsulated mRNA-RBD Vaccine Against SARS-CoV-2 and Its Variants.” We appreciate your useful comments, and suggestions that help us to improve the manuscript.
In the manuscript, first, we have accepted all the changes that were made for the previous revision, and then the “Track Changes” function was used for all the newly made modifications. Additionally, we have corrected some misspelled errors and re-numbered the references accordingly as we had to cite several additional articles. Please find below a point-by-point reply. We hope you will find our responses satisfactory.
Sincerely,
Authors
Comments and Suggestions for Authors:
Thanks for the corrections. I still have minor issues and recommendations as listed below:
Comment 1:
Line 40: «Remarkably, the receptor-binding domain (RBD) induces the specific antibodies». No. The whole spike induces specific antibodies. Those against RBD may give the best neutralization
Response: Thank you for your remark. We have modified the mentioned sentence (lines 40-41):
Remarkably, the receptor-binding domain (RBD) induces a more focused immune response and antibodies that may elicit better neutralization which reduces the risk of antibody-dependent enhancement (ADE) of infection compared with the S-protein [6,7].
Comment 2:
Line 49: Ref 12 and 13 are from 2020. Maybe a more recent review on vaccines against COVID-19 can be cited as for example MDPI Cells Oct 2021 https://pubmed.ncbi.nlm.nih.gov/34685696/
Response: We have considered your suggestion and updated references 12 and 13 with the suggested article (line 49).
Comment 3:
Line 58 “Since the endosome escape efficiency of administered mRNA”
Response: The sentence has been modified accordingly (line 56).
Comment 4:
Line 60 “for the mRNA vaccine is ” “for the synthetic mRNA technologies is ”
Response: We have made this correction (line 58).
Comment 5:
Line 63: “A suitable deliv-” “An optimized deliv-”
Response: We have modified the sentence as suggested (line 60).
Comment 6:
In the introduction is missing a line describing BNT162b1 and reference Dec 2020 https://pubmed.ncbi.nlm.nih.gov/33053279/
Response: We have cited the suggested publication in the introduction (line 54).
Comment 7:
Line 91: could the full mRNA sequence (and the resulting protein sequence) be disclosed in a supporting figure?
Response: We are disclosing supplementary file 1 containing the protein amino acid sequence of our mRNA vaccine product. In the manuscript, it is cited as supplementary file 1 (line 186). However, we would like to leave the full mRNA sequence with optimized codons private.
Comment 8:
Figure 1H: how long was the formulation stored at those temperatures in 1h?
Response: The encapsulation efficiency was measured at temperatures 4 ℃, -20 ℃, and -80 ℃ for 1, 7, and 14 days. Figure 1H represents the encapsulation efficiency for 14 days. For making the figure more understandable we have also included this information in the legend of figure 1H (line 205).
Comment 9:
Line 224. It should be clearly said here that it is N1-Methylpseudouridine (m1Ψ) i
Response: The change has been made (line 228).
Comment 10:
Line 236 “Hence, m1Ψ triphosphate was” “Hence, m1Ψ was”
Response: The change has been made accordingly (line 232).
Comment 11:
Figure 5: I do not understand in A that CD4 and CD8 pos cells are not clearly CD3 positive. Or we do not see here CD3 negative cells in the graph (and they are CD3 positives although they are in the lower part of the dot plots?)?
Response: We apologize for the ambiguity caused. In order to make the figure clearer, we have changed the scale of the Y-axis to logarithmic scaling and the gating strategy. Please see Figure 5A.
Comment 12:
In the discussion (for example line 314) the differences between BNT162b1 and the present work should be mentioned as did the authors in their answer “Firstly, the sucrose containing storage buffer of mRNA-LNP in our study makes it more stable than BNT162b1 at 2-8°C. Secondly, the components of LNP in our study differs from the LNP of Pfizer-BioNTech. The stability of BNT162b1 at 2–8 °C is up to 5 days. Our mRNA-LNP was stable at 4°C for up to 14 days. Most importantly, our study indicates the third boost immunization maybe more helpful in fighting against Delta variant strain than against Omicron. The basic structure of mRNA comprises (i) protein encoding open reading frame (ORF), (ii) 5′ and 3′ UTRs, (iii) a 7-methyl guanosine 5′ cap structure, and (iv) a 3′poly(A) tail. These components can be modified or altered to increase the stability, translation efficiency, and immune-stimulatory feature of mRNA. The 5′ UTR, 3′ UTR, and poly (A) tail elements profoundly influence the stability and translation of mRNA, both of which are critical concerns for vaccines. Compared to BNT162b1, we selected 5' and 3' UTRs of β‐globin of Xenopus laevis. 5' and 3' UTRs of β‐globin of Xenopus laevis are found to enhance the translation and stability of mRNA in many studies (Linares-Fernández et al. 2021 (doi.org/10.1016/j.omtn.2021.10.007); Schlake et al. 2012 (doi.org/10.4161/rna.22269)). Poly(A) tails of 100 nt are ideal for designing mRNA vaccines because poly(A) tail length affects mRNA decay by modulating 3′ exonucleolytic degradation. So we selected 100 nucleotide-long poly(A) tail. Additionally, we selected the tissue plasminogen activator (tPA) signal sequence as the signal peptide. We introduced the tPA signal sequence in order to increase the RBD antigen expression and secretion.”
Response: We have taken your suggestion into consideration and included this information in the discussion part. The new references have been also added accordingly (lines 319-333):
Moreover, several advantages of the vaccine in this study deserve to be mentioned. The sucrose-containing storage buffer of mRNA-LNP in our study makes it relatively stable at 2-8 ℃ compared with BNT162b1 [45]. The mRNA-LNP used in this study elicited stability at 4 ℃ for up to 14 days. The basic structure of mRNA comprises a protein-encoding open reading frame (ORF), 5′ and 3′ UTRs, a 7-methyl guanosine 5′ cap structure, and a 3′ poly-A tail. These components can be modified to increase the stability, translation efficiency, and immune-stimulatory feature of mRNA. The 5′ UTR, 3′ UTR, and poly (A) tail profoundly influence the stability and translation of mRNA, both of which are critical concerns for vaccine development. Thus, we selected 5' and 3' UTRs of β‐globin of Xenopus laevis that are found to enhance the translation and stability of mRNA [46,47]. 100 nt-length Poly-A tail is optimal for designing mRNA vaccines as its length influences the decay of mRNA via modulating 3′ exonucleolytic degradation [48]. Hence, 100 nucleotide-long poly-A tail was employed in this study. Additionally, tPA signal sequence was used as the signal peptide in order to enhance the RBD expression and secretion.
